# Electrophysiology-Guided Genetic Characterisation Maximises Molecular Diagnosis in an Irish Paediatric Inherited Retinal Degeneration Population

**DOI:** 10.3390/genes13040615

**Published:** 2022-03-29

**Authors:** Julia Zhu, Kirk A. J. Stephenson, Adrian Dockery, Jacqueline Turner, James J. O’Byrne, Susan Fitzsimon, G. Jane Farrar, D. Ian Flitcroft, David J. Keegan

**Affiliations:** 1Mater Clinical Ophthalmic Genetics Unit, The Mater Misericordiae University Hospital, D07 R2WY Dublin, Ireland; kirkstephenson@hotmail.com (K.A.J.S.); jturner@mater.ie (J.T.); jamesobyrne@mater.ie (J.J.O.); dkeegan@mater.ie (D.J.K.); 2Ophthalmology Department, Children’s University Hospital, Temple Street, D01 XD99 Dublin, Ireland; susan@fitzsimon.ie (S.F.); ian@flitcroft.com (D.I.F.); 3Next Generation Sequencing Laboratory, Pathology Department, The Mater Misericordiae University Hospital, D07 R2WY Dublin, Ireland; adriandockery@mater.ie; 4The School of Genetics & Microbiology, Trinity College Dublin, D02 PN40 Dublin, Ireland; jane.farrar@tcd.ie

**Keywords:** inherited retinal degenerations, paediatric ophthalmology, inherited blindness, retinitis pigmentosa, Leber congenital amaurosis, achromatopsia, retinal dystrophy, panel-based next generation sequencing

## Abstract

Inherited retinal degenerations (IRDs) account for over one third of the underlying causes of blindness in the paediatric population. Patients with IRDs often experience long delays prior to reaching a definitive diagnosis. Children attending a tertiary care paediatric ophthalmology department with phenotypic (i.e., clinical and/or electrophysiologic) evidence suggestive of IRD were contacted for genetic testing during the SARS-CoV-2-19 pandemic using a “telegenetics” approach. Genetic testing approach was panel-based next generation sequencing (351 genes) via a commercial laboratory (Blueprint Genetics, Helsinki, Finland). Of 70 patient samples from 57 pedigrees undergoing genetic testing, a causative genetic variant(s) was detected for 60 patients (85.7%) from 47 (82.5%) pedigrees. Of the 60 genetically resolved IRD patients, 5% (*n* = 3) are eligible for approved therapies (*RPE65*) and 38.3% (*n* = 23) are eligible for clinical trial-based gene therapies including *CEP290* (*n* = 2), *CNGA3* (*n* = 3), *CNGB3* (*n* = 6), *RPGR* (*n* = 5) and *RS1* (*n* = 7). The early introduction of genetic testing in the diagnostic/care pathway for children with IRDs is critical for genetic counselling of these families prior to upcoming gene therapy trials. Herein, we describe the pathway used, the clinical and genetic findings, and the therapeutic implications of the first systematic coordinated round of genetic testing of a paediatric IRD cohort in Ireland.

## 1. Introduction

Inherited retinal degenerations (IRDs) account for over one third of the underlying causes of blindness in the paediatric population [1,2,3,4,5,6,7,8]. IRDs are a clinically and genetically inhomogeneous group of progressive blinding genetic diseases that can present at any stage from birth through to late middle age. There are over 300 genes identified to date that are associated with isolated or syndromic IRDs [9,10] making accurate molecular diagnosis challenging. Patients with IRDs often experience long delays seeing an average of eight clinicians over 7 years prior to reaching a definitive diagnosis [11,12,13]. The current standard of care necessitates a harmonised clinical and genetic diagnosis to access novel disease-modifying treatments [14].

Clinical diagnoses can be classified according to various features [15,16] including (a) cell type (i.e., cone vs. rod vs. combined dysfunction); (b) primarily targeted retinal region (i.e., macula vs. periphery vs. panretinal); (c) age of onset (i.e., congenital vs. juvenile vs. adult-onset); (d) severity of visual dysfunction (e.g., mild vs. moderate vs. severe); (e) cadence (i.e., stationary vs. progressive); and (f) absence or presence of extra-ocular clinical involvement (i.e., isolated vs. syndromic forms). Non-retinal ophthalmic features include juvenile cataract [17,18,19,20], sensory nystagmus [21] and refractive error [22]. Syndromic IRDs may involve other sensory deficits (e.g., sensorineural hearing loss and retinitis pigmentosa (RP) in up to 18% of children with IRD, i.e., Usher syndrome, OMIM#276900) or require physician management for other body systems (e.g., diabetes mellitus or renal disease in Bardet–Biedl syndrome, OMIM#209900) [23,24,25].

The historical mainstay of IRD treatment has been predominantly supportive (i.e., correction of refractive error, mobility support, educational support) [26,27,28,29]. With the approval of Luxturna^TM^ (Novartis AG, Basel, Switzerland) [30,31], the first gene therapy for IRD (biallelic *RPE65*-associated retinopathy), approved by the US Food and Drug Administration and European Medicines Agency, the relevance of clarifying the genetic aetiology of paediatric (and adult) IRD has been substantiated. There are multiple gene therapy clinical trials underway or planned for other IRD conditions [19,32,33,34,35,36,37,38,39,40] including choroideraemia (*CHM*, OMIM*300390), achromatopsia (*CNGA3*, OMIM*60053; *CNGB3*, OMIM*605080), non-syndromic RP (*RPGR*, OMIM*312610; *MERTK*, OMIM*604705), X-linked retinoschisis (*RS1*, OMIM*300839), Leber congenital amaurosis (LCA, *CEP290*, OMIM*610142) and Usher syndrome (*MYO7A*, OMIM*276903). In childhood, many ‘stationary’ IRDs show preservation of retinal architecture, while some ‘progressive’ IRDs have not yet resulted in significant retinal destruction/atrophy, providing a window of opportunity for intervention (e.g., gene therapy) [33,41,42,43,44]. Thus, early confirmation of genotype is critical to gain access to these disease-modifying therapies. Genetic diagnosis also aids family risk determination where families may yet be incomplete so pedigrees with genetically determined IRD can be appropriately evaluated, investigated, and counselled.

To date, the majority of patients recruited to the Irish national IRD programme (Target 5000) have been adults [14,45,46,47]. Herein, we describe the pathway used to identify and genetically investigate a cohort of children and adolescents with IRD attending a tertiary care paediatric ophthalmology department, the clinical and genetic outcomes, and the therapeutic implications of the first systematic coordinated round of paediatric genetic testing for IRDs in Ireland.

## 2. Materials and Methods

The database of all patients (*n* = 1045) undergoing electrophysiology (EP, full field electroretinography (ERG), multifocal ERG, and/or visually evoked potentials (VEP)) at a tertiary-referral paediatric ophthalmology department (Children’s University Hospital, Temple Street, Dublin, Ireland) between February 2006 and April 2020 were screened for features consistent with IRD. A total of 87 reports (8.3%) consistent with IRDs were identified and a retrospective chart review was conducted to confirm a clinical phenotype of IRD.

Due to limitations from the SARS-CoV-2-19 pandemic, a “telegenetics” approach was adopted to reduce repeated hospital exposures, where possible. Patients or parents were contacted by phone to obtain family history and to consent for genetic testing. All parents/guardians and/or patients (if meeting assent criteria) provided informed consent/assent for genetic testing. This study was approved by the institutional review board of Mater Misericordiae University Hospital, Dublin, Ireland, and abides by the tenets of the Declaration of Helsinki. A saliva sample kit for genetic analysis (Oragene DNA OG-500/OGD-500, DNA GenoTek Inc., Ottawa, ON, Canada) was sent directly to the patient’s home address to minimise additional hospital exposures. A panel-based next generation sequencing (pNGS) analysis of 351 IRD-implicated genes (nuclear and mitochondrial genome) was performed by an accredited laboratory (Blueprint Genetics, Helsinki, Finland) [48]. All relevant variants were reported in HGNC nomenclature, compared against a reference genome (GRCH37/HG19), and confirmed with Sanger sequencing as per the commercial laboratory’s protocols.

Findings from pNGS were discussed in a multidisciplinary team (MDT) setting comprising ophthalmologists, clinical and molecular geneticists, and genetic counsellors to ensure correlation of the phenotype with genotype prior to feedback of genetic results to the patients/parents by the genetic counsellor.

Assessed clinical data included demographics, age of onset, ophthalmic and systemic symptoms/diagnoses, family history, and comprehensive ophthalmic examination (i.e., LogMAR visual acuity (VA), cycloplegic retinoscopy, and dilated slit lamp biomicroscopy and/or indirect ophthalmoscopy). Where possible, colour fundus photography, fundus autofluorescence and optical coherence tomography were captured to further characterise and stage the retinal phenotype. In cases of syndromic IRD, relevant input from other expert medical specialities (e.g., neurodevelopmental, metabolic, cardiology, renal, etc.) was sought.

## 3. Results

### 3.1. Demographics

A total of 87 individuals were identified as having electrophysiology features consistent with IRD (e.g., rod and/or cone photoreceptor dysfunction). See Table 1 and Figure 1. Out of the 87 individuals, 70 patients (80.5%) from 57 pedigrees agreed to or were available for genetic testing. Mean patient age at time of testing was 13.94 years, with 64.3% male and 35.7% female.

### 3.2. Visual Acuity

Quantifiable VA measurements were possible in 88.6% (62/70) of cases (i.e., children able to reliably respond to LogMAR letters/pictures), with the majority of children who were unable to give formal VA readings having a diagnosis of LCA (62.5%, 5/8). The most severe visual disturbance was noted in patients with LCA, Stargardt disease (STGD) and cone dystrophy (CD). Table 2 and Figure 2 provides a more detailed analysis. The variable VA noted in individuals with LCA, STGD, CD, and RCD may be attributable to severity/stage of disease associated with differing ages and VA is not always the most reliable marker of disease severity on IRD (e.g., early visual field loss in RP).

### 3.3. Refractive Error

Refractive error was recorded for 75.7% of cases (Table 3). High myopia is strongly associated with congenital stationary night blindness (CSNB) which is in keeping with the literature [22,49]. Astigmatism was also greatest in CSNB. A total of 41.5% of patients had ≥6D ametropia making refractive error the most common treatable feature in paediatric IRD.

### 3.4. Other Ocular Co-Morbidities

No patients had keratoconus or glaucoma. Five patients had evidence of cataract; one patient with ACHM had Mittendorf dot cataract and three patients (16.7%) with RCD had bilateral posterior subcapsular cataract at a mean age of 18.5 years. One patient with XLRS had history of unilateral rhegmatogenous retinal detachment (RRD) repair (pars plana vitrectomy with scleral buckle) and subsequently underwent cataract surgery at age 16. RRD was detected in only one eye of this total paediatric IRD cohort (0.7% of eyes or 1.4% of patients, all diagnoses). One patient with BVMD had evidence of choroidal neovascular complex in the right eye requiring intravitreal anti-vascular endothelial growth factor injection with good visual result. Cystic macular lesions (CML) were noted in all cases with XLRS (*n* = 7) [50]. No patients with non-XLRS diagnoses had CML though detection of this finding may have been impacted by limited facility for detailed multimodal imaging (MMI, i.e., optical coherence tomography, OCT) in young patients. A total of 11 patients (15.9%) had amblyopia, calculated using a ≥2-line interocular VA discrepancy, though this does not account for the bilateral stimulus deprivation amblyopia of severe early onset IRD affecting VA. Table 4 provides a more detailed analysis.

### 3.5. Genetic Analysis

Of 70 patient samples from 57 pedigrees undergoing pNGS, causative genetic variants were detected for 60 patients (85.7%) from 47 pedigrees (82.5%). The most common genetic aetiologies were *RS1* (11.7%, *n* = 7), *CNGB3* (10%, *n* = 6), *ABCA4* (8.3%, *n* = 5), *RPGR* (8.3%, *n* = 5), and *NYX* (6.7%, *n* = 4). See Table 5.

In total, 63 variants in 27 IRD-associated genes were identified with 90.5% (*n* = 57) being pathogenic (American College of Medical Genetics, ACMG class 5, 65.1%, *n* = 41) or likely pathogenic (ACMG class 4, 25.4%, *n* = 16) variants and only 9.5% (*n* = 6) being variants of unknown significance (VUS, ACMG class 3) requiring further work to assess degree of deleteriousness on protein function (e.g., in silico analysis) or segregation analysis. There were 11 novel variants (17.5%) detected (Appendix A).

The most common inheritance pattern for genetically resolved cases was autosomal recessive (AR, 55%, *n* = 33) with X-linked recessive (XL, 30%, *n* = 18) and autosomal dominant (AD, 15%, *n* = 9) less frequent. A total of 33.3% (*n* = 11) of recessive disease was due to homozygous mutations, with the remainder explained by compound heterozygous variants in the relevant gene. Of 10 unresolved patients with clinically AR IRD, a single allele was detected in 5 patients (50%, ACMG class 5 in two cases, class 4 in one case, and class 3 in two cases). Further studies including whole gene sequencing (i.e., single allele cases) and/or whole exome/genome sequencing (i.e., no candidate genetic variants) are planned to resolve these cases [51].

Of the 60 genetically resolved IRD patients, 5% (*n* = 3) are eligible for approved therapies (i.e., Luxturna™, *RPE65*) and 38.3% (*n* = 23) are eligible for clinical trial-based gene therapies including *CEP290* (*n* = 2), *CNGA3* (*n* = 3), *CNGB3* (*n* = 6), *RPGR* (*n* = 5), and *RS1* (*n* = 7).

pNGS revealed additional findings in ocular-implicated genes in 33 patients with 15 patients having more than one additional variant identified (Appendix A); however, these variants were mainly ACMG class 3 (VUS) and did not correlate well with the clinical phenotype. As whole exome testing was not conducted, secondary findings (i.e., reportable non-retinal genes of significance to ongoing patient welfare) were not included in the analysis [52,53].

## 4. Discussion

This is the first systematic coordinated genetic assessment of a paediatric IRD cohort in Ireland which resulted in an 85.7% genetic resolution rate from first-tier pNGS testing (with 90.5% of variants classified as pathogenic or likely pathogenic), superior to our adult cohort (~70% genetically resolved) [45] and other pNGS IRD studies [15,20,54,55,56,57,58,59]. Similar positive correlation between paediatric age group and likelihood of obtaining genetic diagnosis has been previously noted [56,57]. This is likely due to (1) selective genetic testing only of patients with abnormal EP and (2) lack of advanced acquired disease mimicking IRD (as seen in adult populations) [50].

In childhood-onset IRD, some phenotypes are specific for mutations in a single gene or a relatively small subset of genes, e.g., XLRS (*RS1*), LCA (including *RPE65, CEP290, AILP1, RDH12, CRX* and *CRB1*, among others), and CSNB (including *NYX*, *TRPM1* and *CACNA1F*, among others) [53]. EP may be particularly helpful in refining the clinical diagnosis, thus narrowing the scope of the genetic spectrum of interest, e.g., 1., electronegative ERG in a male with macular retinoschisis suggests *RS1* genotype, while, e.g., 2., CSNB can be categorised by EP subtypes (i.e., Riggs vs. Schubert–Bornschein types). In patients with broader, less specific clinical diagnoses (e.g., retinal dystrophy or rod-cone dystrophy), a larger list of potential genetic aetiologies must be considered (i.e., >100). Thus, there is a lower chance of receiving a molecular diagnosis and greater chance of uncovering misleading non-causative variants in other IRD-associated genes [56]; it is these cases in which pNGS is most effective. In our cohort, 81.3% of patients diagnosed with RCD and 70.6% of patients diagnosed with CD (excluding STGD) received a conclusive genetic diagnosis, demonstrating the power of a broader genetic scope (i.e., pNGS of 351 genes in this case) in resolving heritable diseases with heterogeneous genetic makeup.

Associated ophthalmic features (e.g., cataract, CML) were found in a minority of patients in this cohort (14.3%). This figure may increase as patients enter adulthood (i.e., later stage of the disease process) as detection/monitoring of subtle findings (i.e., CML) may improve with greater cooperation with detailed examination (e.g., nystagmus, photophobia, fundus contact lens) and MMI (e.g., OCT) [50]. Although refractive error may increase with age in healthy eyes [60], in IRD, retinal feedback on eye growth is impaired due to defocus and the inherent retinal dysfunction, which may accelerate axial elongation of the globe [48,49]. Although likely an adaptive strategy to maximise vision, this may lead to acquired sequelae including angle closure (high hyperopia) and retinal breaks, detachment and/or atrophy (high myopia) [61,62]. Uncorrected refractive errors can cause ametropic amblyopia of their own accord in addition to the underlying retinal dysfunction of IRD and thus core measures such as accurate refractive correction and amblyopia treatment (e.g., patching) remain the cornerstone of care to prevent acquired visual loss in children with IRD.

In a paediatric cohort, early genetic diagnosis empowers patients and their families to make decisions regarding their care. Genetically guided prognosis allows educational, employment, and supportive planning for the long term. Ocular supports such as refraction, low vision aids, and mobility training can be introduced early to maximise function. In scenarios where young families may yet be incomplete, genetic counselling is particularly important in making informed reproductive decisions [61].

The need to achieve an accurate genetic diagnosis for IRD patients is increasingly important as novel gene therapies are in clinical trials for a growing number of aetiologies [32,34,51,63] and a molecular diagnosis is often a prerequisite for access to clinical trials and approved treatments. In this cohort, 43.3% of the children (26 out of 60 tested positive) meet the criteria for gene therapies (approved or in clinical trials). Although long term efficacy from once-off gene therapy for IRD is not known [64], access to disease-modifying treatments prior to the onset of structural damage to the retina (i.e., photoreceptor/RPE atrophy) will likely become the standard of care for IRDs along with refractive correction and supportive measures. This research group has plans to participate in upcoming phase I/II/III gene therapy clinical trials for *RPGR, CNGB3*, and *USH2A* which paediatric (and adult) patients from these pedigrees may directly benefit from.

### Limitations

This study was carried out during the global SARS-CoV-2-2019 pandemic and thus some patients/parents were unwilling/unable to travel for in-person clinical re-assessment. Therefore, some cases (*n* = 17) were excluded and phenotypic information is not current for all cases. This also limited opportunities to perform MMI on all patients, though the younger of the cohorts may not have been amenable regardless. Alternate clues and techniques must be employed, including modified EP testing setups more tolerable/appropriate for young children [65]. Limitations of this study also include its retrospective nature and the low number of patients, an issue with many observational studies of rare diseases. However, we feel that this study represents the paediatric IRD population in Ireland, capturing ~27% of the estimated 365 Irish population ≤18 years with IRD. Only patients having been referred for electrophysiology were screened and thus patients with more severe ocular or syndromic (i.e., intellectual disability) phenotypes may have been excluded due to electrophysiology non-completion. Inherited vitreoretinopathies, which are largely monogenic (e.g., Stickler syndrome, OMIM#108300) were not captured by the EP screening approach described here as frank rod and/or cone dystrophy would not be apparent. This may bias the demographic data output from this study, describing a certain subset of IRDs. Definitive statements are difficult to make with small numbers of genetically heterogeneous diseases; however, the data herein support the use of an EP-guided clinically validated pNGS approach to the investigation of paediatric IRD to maximise positive gene detection.

Further work is ongoing to assess an additional cohort of the paediatric population with inconclusive EP findings (excluded in the first round) suspected to have IRD. This includes repeat EP and clinical assessment prior to genetic testing (pNGS).

## 5. Conclusions

This study represents a vital initial step in the genetic characterisation of children with IRD to empower them with prognosis, genetic counselling (family planning tools/evidence), and access to novel therapies, where available. Visual electrophysiology is a useful objective adjunct to history, examination, and MMI to clinically diagnose IRD in children who may have difficulty completing a full battery of typical phenotypic characterisation tests. The early introduction of genetic testing in the diagnostic/care pathway for children with phenotypic (i.e., clinical and/or electrophysiologic) findings suggestive of IRD is critical for genetic counselling of these families prior to upcoming gene therapy trials.

## Figures and Tables

**Figure 1 genes-13-00615-f001:**
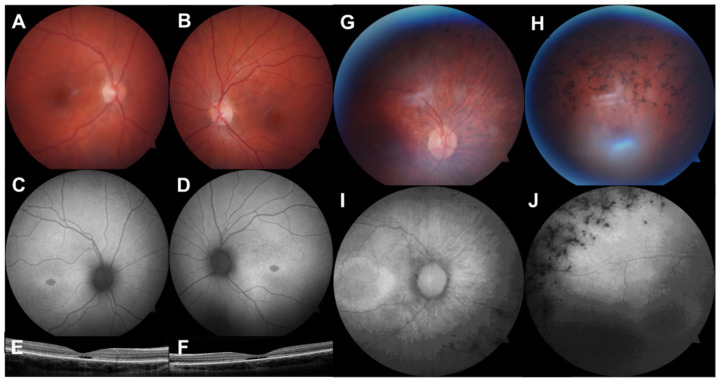
Multimodal retinal imaging of 2 paediatric IRD cases. (**A**,**B**). Colour fundus photographs (FF450 plus, Carl Zeiss MediTec, Dublin, CA, USA) showing an altered foveal reflex. (**C**,**D**). Fundus autofluorescence images showing symmetrical hypoautofluorescence at the fovea with a hyperautofluorescent margin. (**E**,**F**). Optical coherence tomography (iVue 80, Optovue Inc., Fremont, CA, USA) showing outer retinal cavitation at the fovea. This 13-year-old female had Stargardt disease due to compound heterozygous pathogenic *ABCA4* variants. (**G**,**H**). Colour fundus photographs of the right eye showing optic nerve head pallor, arteriolar attenuation, and intraretinal pigment migration. (**I**,**J**). Fundus autofluorescence showing a hyperautofluorescent macular ring and peripheral hypoautofluorescence. This 10-year-old female had autosomal dominant RP due to a likely pathogenic *PRPF8* variant.

**Figure 2 genes-13-00615-f002:**
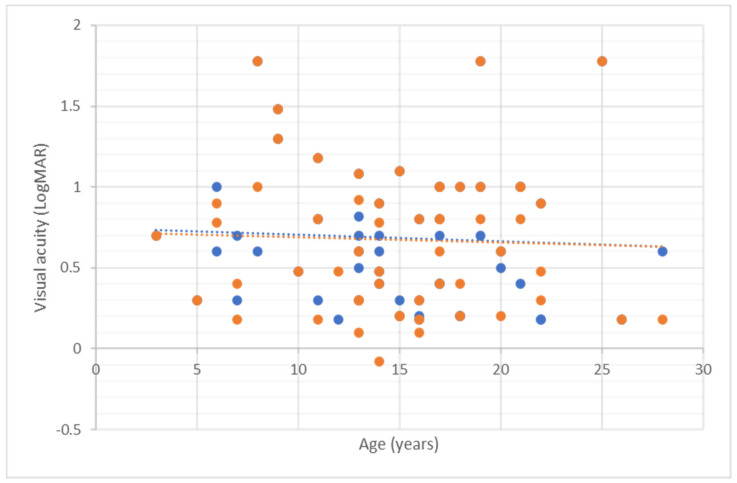
Mean visual acuity against age (in years) for all patients. Blue dots represent the right eye and orange dots represent the left eye.

**Table 1 genes-13-00615-t001:** Demographic information by clinical diagnosis (in alphabetical order).

Diagnosis	N = (%)	Age (Range)	SD Age	Gender % Male	Referring EP Findings
AS	1 (1.4)	9	-	100%	Flat ERG (i.e., no retinal response)
BVMD	3 (4.3)	12.67 (5–18)	6.43	67%	EOG: reduced Arden ratio
CD	17 (24.3)	13.94 (2–19)	5.55	53%	ERG: Unrecordable cone responses without rod involvement
CSNB	9 (12.9)	18 (7–28)	6.69	67%	ERG: Electronegativity, normal 30 Hz waveform, with residual rod response
LCA	10 (14.3)	5 (1–9)	4.16	70%	ERG: No recordable retinal function VEP: Residual small amplitude flash VEP response
RCD	18 (25.7)	16.11 (6–22)	4.54	56%	ERG: Reduced/absent rod response with less pronounced cone attenuation
STGD	5 (7.1)	19.6 (14–25)	4.72	60%	ERG: Reduced 30 Hz flicker with preservation of rod function
XLRS	7 (10)	13.14 (7–21)	4.43	100%	ERG: Electronegativity

AS−Alström syndrome. BVMD−Best vitelliform macular dystrophy. CD−cone dystrophy. CSNB−congenital stationary night blindness. EOG−electro-oculogram. ERG−electroretinogram. LCA−Leber congenital amaurosis. RCD−rod cone dystrophy. STGD−Stargardt disease. VEP−visually evoked potential. XLRS−X-linked retinoschisis.

**Table 2 genes-13-00615-t002:** Mean visual acuity (VA) in LogMAR by clinical diagnosis (in alphabetical order). Patients not accounted for in this table were unable to provide formal VA assessment, predominantly LCA patients who were too young for formal VA assessment.

Diagnosis	No. of Patients (N =)	VA Right Eye	VA Left Eye
AS	1/1	1.48	1.48
BVMD	3/3	0.10 ± 0.17	0.10 ± 0.17
CD	15/17	0.89 ± 0.39	0.92 ± 0.34
CSNB	9/9	0.37 ± 0.19	0.36 ± 0.20
LCA	5/10	0.94 ± 0.60	0.98 ± 0.63
RCD	17/18	0.36 ± 0.34	0.36 ± 0.34
STGD	5/5	1.10 ± 0.41	1.11 ± 0.39
XLRS	7/7	0.48 ± 0.11	0.38 ± 0.26

AS−Alström syndrome. BVMD−Best vitelliform macular dystrophy. CD−cone dystrophy. CSNB−congenital stationary night blindness. LCA−Leber congenital amaurosis. RCD−rod cone dystrophy. STGD−Stargardt disease. XLRS−X-linked retinoschisis.

**Table 3 genes-13-00615-t003:** Mean refraction and maximum cylinder in dioptres (D) for both eyes with standard deviation by clinical diagnosis (in alphabetical order).

Diagnosis	No. of Patients (N =)	Refraction (Spherical Equivalent, D)	Astigmatism (D)
AS	1/1	+7.63	1.75
BVMD	1/3	+2.38	0.25 ± 0.35
CD	16/17	−0.74 ± 6.54	0.92 ± 0.99
CSNB	9/9	−8.99 ± 3.23	2.24 ± 0.99
LCA	7/10	+5.38 ± 2.37	0.82 ± 0.77
RCD	14/18	−4.96 ± 6.45	1.77 ± 1.01
STGD	2/5	−2.91 ± 1.64	1.69 ± 1.10
XLRS	3/7	+2.94 ± 2.32	0.46 ± 0.60

AS−Alström syndrome. BVMD−Best vitelliform macular dystrophy. CD−cone dystrophy. CSNB−congenital stationary night blindness. LCA−Leber congenital amaurosis. RCD−rod cone dystrophy. STGD−Stargardt disease. XLRS−X-linked retinoschisis.

**Table 4 genes-13-00615-t004:** Other ocular co-morbidities by clinical diagnosis (in alphabetical order).

Diagnosis	N=	Cataract (*n* =)	Pseudophakia (*n* =)	CML (*n* =)	RRD (*n* =)	Amblyopia (*n* =)	Glaucoma (*n* =)
AS	1	0	0	0	0	0	0
BVMD	3	0	0	0	0	0	0
CD	17	1	0	0	0	1	0
CSNB	9	0	0	0	0	4	0
LCA	10	0	0	0	0	1	0
RCD	18	3	0	0	0	2	0
STGD	5	0	0	0	0	0	0
XLRS	7	1	1	7	1	3	0
Total	70	5 (7.2%)	1 (1.4%)	7 (10%)	1 (1.4%)	11 (15.7%)	0 (0%)

AS−Alström syndrome. BVMD−Best vitelliform macular dystrophy. CD−cone dystrophy. CML−Cystoid macular lesions. CSNB−congenital stationary night blindness. LCA−Leber congenital amaurosis. RCD−rod cone dystrophy. RRD−Rhegmatogenous retinal detachment. STGD−Stargardt disease. XLRS−X-linked retinoschisis.

**Table 5 genes-13-00615-t005:** Percentage of genetically resolved cases and associated genes from this paediatric IRD cohort within each clinical diagnosis category.

Diagnosis	N =	Genetically Solved %	Associated Gene
AS	1/1	100%	*ALMS1*
BVMD	3/3	100%	*BEST1*
CD	12/17	70.6%	*CNGB3* (*n* = 6), *CNGA3* (*n* = 3), *PDE6H* (*n* = 1), *KCNV2* (*n* = 1)
CSNB	8/9	88.9%	*NYX* (*n* = 4), *TRPM1* (*n* = 3), *CACNA1F* (*n* = 1)
LCA	9/10	90%	*RPE65* (*n* = 2), *CEP290* (*n* = 2), *AIPL1* (*n* = 2), *RDH12* (*n* = 1), *CRX* (*n* = 1), *CRB1* (*n* = 1)
RCD	15/18	83.3%	XL: *RPGR* (*n* = 5), *RP2* (*n* = 1) AD: *PRPF31* (*n* = 2), *RHO* (*n* = 1), *PRPF8* (*n* = 1), AR: *CNGB1* (*n* = 1), *RLBP1* (*n* = 1), *IMPDH1* (*n* = 1), *MYO7A* (*n* = 1), *BBS1* (*n* = 1)
STGD	5/5	100%	*ABCA4*
XLRS	7/7	100%	*RS1*

AD−autosomal dominant. AR−autosomal recessive. AS−Alström syndrome. BVMD−Best vitelliform macular dystrophy. CD−cone dystrophy. CSNB—congenital stationary night blindness. LCA−Leber congenital amaurosis. RCD−rod cone dystrophy. RRD−rhegmatogenous retinal detachment. STGD−Stargardt disease. XLRS−X-linked retinoschisis. XL−X-linked.

## Data Availability

Anonymised source data are available upon reasonable request.

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
