# Peer review of "Electrophysiology-Guided Genetic Characterisation Maximises Molecular Diagnosis in an Irish Paediatric Inherited Retinal Degeneration Population"

_genes, 2022, doi:10.3390/genes13040615_

Round 1
Reviewer 1 Report
This manuscript by Julia Zhu and colleagues is an original, substantial, high-quality, and exceptionally well-written paper. It makes an important contribution to the understanding of the spectrum of gene mutations in the pediatric IRD population.
I only have some minor comments:
- The authors do not state whether NGS was performed only for the affected subjects or as trio analysis. I guess that only the affected subjects were analyzed since the authors state in lines 197-199 that “…variants of unknown significance requiring further work (…) or segregation analysis.” For completeness, Supplementary Table 1 should be supplemented with information regarding the phase of alleles. However, there is no need to adjust the molecular diagnostic rate with respect to phase.
- Please give RefSeq numbers in Supplementary Tables 1 and 2.
- In Supplementary Table 1, some mutations lack the parentheses around the predicted amino acid exchange. I am unsure if this should be an indication that the respective mutation has been functionally studied? If so, this should be stated in the table legend.
- Mostly, the symbol “*” is used in Supplementary Table 1 in the nomenclature of frameshift variants, but sometimes the three letter code “Ter” is used. This should be unified.
Author Response
- The authors do not state whether NGS was performed only for the affected subjects or as trio analysis. I guess that only the affected subjects were analyzed since the authors state in lines 197-199 that “…variants of unknown significance requiring further work (…) or segregation analysis.” For completeness, Supplementary Table 1 should be supplemented with information regarding the phase of alleles. However, there is no need to adjust the molecular diagnostic rate with respect to phase.
Yes, you are correct in that first line NGS was just of IRD-affected children without necessarily phase testing, though some pedigrees with multiple affected siblings were included. We are currently in process of carrying out the phase testing on the parents on all recessive cases. These changes have been added to supplementary table 1.
- Please give RefSeq numbers in Supplementary Tables 1 and 2.
These have been added to Supplementary Tables 1 and 2. Thank you.
- In Supplementary Table 1, some mutations lack the parentheses around the predicted amino acid exchange. I am unsure if this should be an indication that the respective mutation has been functionally studied? If so, this should be stated in the table legend.
Thank you for pointing this out. This is not an indication of functional validation. It is a small difference in style of annotation, all variants have now been amended to include their protein descriptions in parentheses.
- Mostly, the symbol “*” is used in Supplementary Table 1 in the nomenclature of frameshift variants, but sometimes the three letter code “Ter” is used. This should be unified.
Thank you for your comment. We have harmonized the nomenclature and used the “*” symbol as the indicator of a premature termination variant.
Reviewer 2 Report
This was interesting study. It adds to the growing evidence in the field that an early molecular diagnosis of IRD improves patient and family care as well as access to treatment. I am curious if the results of this study have changed the practice of caring for pediatric IRD patients at the Center.
Author Response
Thank you for your comment. Yes, this study represents a shift in the approach to managing children with clinical features of IRD.
It has demonstrated the effective interplay of electrophysiological diagnoses coupled with rapid access to genetic testing. We now employ this model.
It has introduced the concept of remote clinical appointments for parents who wish to avail of them.
Most importantly we are now able to provide a genetic diagnosis in >85% of cases and, as gene therapies become available in trial or approved formats, we are able to offer these disease-modifying therapies to them.